# Probing Intrinsic Neural Timescales in EEG with an Information-Theory Inspired Approach: Permutation Entropy Time Delay Estimation (PE-TD)

**DOI:** 10.3390/e25071086

**Published:** 2023-07-19

**Authors:** Andrea Buccellato, Yasir Çatal, Patrizia Bisiacchi, Di Zang, Federico Zilio, Zhe Wang, Zengxin Qi, Ruizhe Zheng, Zeyu Xu, Xuehai Wu, Alessandra Del Felice, Ying Mao, Georg Northoff

**Affiliations:** 1Padova Neuroscience Center, University of Padova, Via Orus 2/B, 35129 Padova, Italy; patrizia.bisiacchi@unipd.it (P.B.); alessandra.delfelice@unipd.it (A.D.F.); 2Department of General Psychology, University of Padova, Via Venezia, 8, 35131 Padova, Italy; 3The Royal’s Institute of Mental Health Research & University of Ottawa, Brain and Mind Research Institute, Centre for Neural Dynamics, Faculty of Medicine, University of Ottawa, 145 Carling Avenue, Rm. 6435, Ottawa, ON K1Z 7K4, Canada; ycata092@uottawa.ca; 4Department of Neurosurgery, Huashan Hospital, Shanghai Medical College, Fudan University, Shanghai 200040, China; zangdizhbjig@126.com (D.Z.); wangzhe@126.com (Z.W.); qizengxin@126.com (Z.Q.); zhengruizhe@126.com (R.Z.); xuzeyu@126.com (Z.X.); wuxuehai@126.com (X.W.); 5Shanghai Key Laboratory of Brain Function and Restoration and Neural Regeneration, Shanghai 200040, China; 6State Key Laboratory of Medical Neurobiology, MOE Frontiers Center for Brain Science, School of Basic Medical Sciences and Institutes of Brain Science, Fudan University, Shanghai 200032, China; 7National Center for Neurological Disorders, Shanghai 200040, China; 8Neurosurgical Institute, Fudan University, Shanghai 200040, China; 9Shanghai Clinical Medical Center of Neurosurgery, Shanghai 200040, China; 10Department of Philosophy, Sociology, Education and Applied Psychology, University of Padova, Piazza Capitaniato, 3, 35139 Padova, Italy; federico.zilio@unipd.it; 11Department of Neuroscience, Section of Neurology, University of Padova, Via Belzoni, 160, 35121 Padova, Italy; 12Mental Health Center, Zhejiang University School of Medicine, Hangzhou 310013, China; 13Centre for Cognition and Brain Disorders, Hangzhou Normal University, Hangzhou 310013, China

**Keywords:** intrinsic neural timescales, electroencephalography, consciousness, permutation entropy, neural time delay

## Abstract

Time delays are a signature of many physical systems, including the brain, and considerably shape their dynamics; moreover, they play a key role in consciousness, as postulated by the temporo-spatial theory of consciousness (TTC). However, they are often not known a priori and need to be estimated from time series. In this study, we propose the use of permutation entropy (PE) to estimate time delays from neural time series as a more robust alternative to the widely used autocorrelation window (ACW). In the first part, we demonstrate the validity of this approach on synthetic neural data, and we show its resistance to regimes of nonstationarity in time series. Mirroring yet another example of comparable behavior between different nonlinear systems, permutation entropy–time delay estimation (PE-TD) is also able to measure intrinsic neural timescales (INTs) (temporal windows of neural activity at rest) from hd-EEG human data; additionally, this replication extends to the abnormal prolongation of INT values in disorders of consciousness (DoCs). Surprisingly, the correlation between ACW-0 and PE-TD decreases in a state-dependent manner when consciousness is lost, hinting at potential different regimes of nonstationarity and nonlinearity in conscious/unconscious states, consistent with many current theoretical frameworks on consciousness. In summary, we demonstrate the validity of PE-TD as a tool to extract relevant time scales from neural data; furthermore, given the divergence between ACW and PE-TD specific to DoC subjects, we hint at its potential use for the characterization of conscious states.

## 1. Introduction

Complex physical systems are characterized by their own intrinsic temporal and spatial dimensions. It follows that a deeper understanding of such dimensions is required to capture the complexity and predict the future behaviors of such systems.

In particular, this temporal dimension might be influenced by one or more sources of delay, which contribute significantly to the system’s temporal structure by generating dominant time scales of activity [1]. Therefore, capturing the essential features of a complex system also requires the inference of these time delays directly from time series data. This task is of utmost relevance, as intrinsic temporal dependence structures characterize many dynamical processes with a vast range of examples, ranging from optics to biology, astronomy, and many others [2].

Immersed in an environment characterized by a diverse set of time scales, the brain is no exception. Growing evidence has shown that the brain displays different preferential temporal durations in both its spontaneous and its task-evoked activities [3,4,5], likely adapting to the temporal durations of external environmental inputs [6,7]. These concepts are summarized in the notion of **intrinsic neural timescales** (**INTs**), which are defined as “temporal windows of neural spontaneous activity during which neural activity is strongly correlated with itself” [8]. Time delays, such as (but not restricted to) INTs, are ubiquitous features of self-organizing dynamic systems such as the brain. However, INTs are not only a signature of self-organized complexity: an additional functional role for processes such as temporal integration and segregation has been recently proposed [3,8,9]. Therefore, accurate estimation of the temporal dependences of neural spontaneous activity—specifically in the form of INTs—might also improve the understanding of behavior and cognition.

Starting with the introduction of the notion of INTs, the dominant timescales of neural activity have been probed by means of different methodologies. The majority of studies of these issues have leveraged the property of autocorrelation function (ACF) to estimate the timescale over a signal that shows periodic patterns [10]: thus the auto-correlation window (ACW) approach, which involves computing the ACF of the signal of interest and the later estimating its fall to its 50% or 0% value [8,9,11,12]. With ACW, various groups have successfully found coherent results across different modalities, such as fMRI, M/EEG, and single-cell recordings [4,9,13,14], and that the relevance of INTs extends to perception and cognition (see [8] for a review).

ACF is a valuable tool that is not exclusive to neuroscience: it is widely used to estimate time delays in complex systems across several academic disciplines [1]. However, one alternative solution to this standard has been recently proposed, which relies on the popular information-theory quantifier developed in [15]: permutation entropy (PE). PE belongs to a larger family of measures that quantify the informational content of an observable phenomenon, which are all rooted in the original formulation Shannon’s entropy [16]. Given the probability distribution $P=\left\{p_{i};i=1,\ldots ,k\right\}$
$$S\left(P\right)=-\sum_{i=1}^{k}p_{i}\ln p_{i}$$

*S* quantifies the degree of uncertainty associated with the probability distribution of the observed phenomenon. The members of this family of information–theoretical measures usually differ from one another with respect to how the process’ probability distribution is inferred from empirical data. Importantly, estimation of the probability distribution of a time-series is problematic. Counting the relative frequency of events [17] (for example, via coarse-graining values and placing them in bins) assumes ergodicity, which is rarely true for systems that have memory (e.g., biological systems [18]). In this way, the ordering of events in the direction of the arrow of time is lost. PE solves these issues with a “symbolization” procedure, which involves decomposing the continuous signal of interest into a series of “motifs”—signal partitions that are ranked into ordinal patterns; based on the relative occurrence of these motifs, a probability distribution is inferred, onto which the Shannon entropy formula can be applied. A brief description of the method is provided in the Methods section (Section 2.1); for a thorough description of PE, which is not the focus of this study, please refer to [15].

In this sense, PE is particularly suited to inferring temporal relations in a time series since its symbolization technique considers temporal patterns. How can PE support the estimation of time delays from time series? Symbolization requires two parameters: the embedding dimension *D*, which controls how many consecutive time points are needed to “build” a symbol/ordinal pattern; and the embedding delay *tau*, which controls the temporal distance between the consecutive time points of a single symbol. If PE is computed multiple times with an increasing *tau*—which is equal to increasing the temporal granularity of the investigation—a particular graph is obtained, which might display one or multiple local minima. Those minima should then correspond to the time delays of the time series: the intuition is that the entropy associated with the observed phenomenon should be minimal when the temporal granularity (*tau*) matches its dominant time scale. An alternative way of conceptualizing this method is that it is directed at capturing the complete duration of events. If *tau* is equal to the period, then each symbol encompasses the whole period, and every symbol is the same. This process results in a probability distribution with maximum precision and therefore minimum entropy.

A series of studies of model systems and real-world systems have confirmed that PE exhibits a minimum value at a *tau* corresponding to the system’s own time scale [19,20,21,21]; however, despite evidence of the efficacy of PE in recovering time scales from time series, this method has yet to be used in a neuroscientific context, in which the estimation of time scales—and particularly the estimation of INTs—is a relevant matter.

The importance of INTs is not restricted to temporal input processing [5,22,23]. In fact, recent evidence has shown that loss of consciousness is consistently followed by an anomalous alteration of INT values [13,14,24], compared to the range of values typically displayed by healthy conscious populations; that this peculiar alteration is observed across different unconscious states (sleep, anesthesia, and disorders of consciousness—DoC [25]) suggest a strong relation to consciousness [26]. In this sense, the temporospatial theory of consciousness (TTC) [27,28] postulates an important role for the temporal dynamics of the brain’s spontaneous activity in shaping the form/context of conscious states. Hence, in the framework of TTC, the estimation of INTs goes even beyond the experimental need to predict future behaviors of a complex system such as the brain and is therefore an important prerequisite to investigating different states of consciousness.

Testing for hypotheses that relate INTs to conscious states requires the cleanest experimental contrast possible. However, unconscious states are notoriously characterized by different regimes of nonstationarity [29,30] and nonlinearity [31,32]: this fact might hinder the experimental methods used to uncover differences exclusively related to consciousness, which often rely on strong assumptions of the nonstationarity and/or nonlinearity of the analyzed signal.

Hence, given the important challenges that neural data—and especially EEG data—pose for the identification of time delays, permutation entropy–time delay estimation (PE-TD) is a promising approach that could aid the advance of many neuroscientific disciplines and is well suited to test for hypotheses relevant to consciousness research.

Our aim for this study is two-fold: (i) the validation and exploration of the use of PE-TD for the estimation of neural time scales, with the aid of synthetic data (when the ground truth is known) and real-world EEG data; and (ii) providing evidence for a parallel role of these measures in exploring the properties of EEG recordings from people with clinical loss of consciousness.

## 2. Materials and Methods

### 2.1. Estimation of EEG Time Scales through Permutation Entropy—PE-TD

Permutation entropy (PE) calculation requires a symbolization procedure: a series of steps that map ordinal patterns into permutation patterns directly from time series data. The symbolization procedure is briefly recapitulated here.

Given a time series *X* = {Xt:t = 1, ⋯, N}, symbolization involves the careful choice of two parameters: the embedding dimension D and the embedding delay *tau* (τ). The first step involves producing vectors of length D, consisting of consecutive time-ordered values: for instance, at D = 3, each vector will consist of three consecutive time points. The embedding delay *tau* controls the temporal distance that separates each consecutive value in the vector: at *tau* = 1, the original time granularity will be preserved; while at *tau* = 2, vectors will consist of every other time point from the original time points, and so on.

Thus, the symbol *S* is constructed as:S_i_ = {X_i_, X_i+τ_, X_i+2τ_, …, X_i+D−1_} for i = 1, 2, …, N

Next, the values in each of vector *S_i_* are ranked in ascending order, and their vector entries are substituted by their rank order, eventually resulting in the formation of an ordinal pattern. Every ordinal pattern will then correspond to a permutation pattern (a “symbol”). The last step involves computing the Shannon entropy of the probability distribution *P*, of which its pi units are the frequencies associated with all the possible permutation patterns extracted with the symbolization procedure:$$PE=-\sum_{i=1}^{D!}p_{i}\;\ln p_{i}$$

#### PE-TD

Fixing the parameter *D* and measuring PE values as functions of the embedding delay tau, one can easily visualize how tau influences the resulting PE values. The most important assumption is that the signal will be more “predictable” if tau matches the timing of the intrinsic time delays of the system: a system that has an intrinsic time delay will have a narrower probability distribution at that particular time scale/tau (and thus, more predictability) since events that happen periodically in a time series will result in the same permutation pattern at that particular temporal grain. This phenomenon, in turn, is associated with the observation of a clear minimum in the PE vs. time delay graph.

We extracted the absolute minimum of the PE vs. time-delay graph for each time series that we analyzed. Time series data, in the particular case of this study, were represented by either synthetic signals or EEG channel data. Hence, we defined the permutation entropy–time delay (PE-TD) estimation of the intrinsic time scale as the corresponding minimum of the PE vs. time delay graph, multiplied by the sampling rate, to convert this estimation into seconds.

One important step when applying this method is the choice of an appropriate upper limit for tau: setting a low upper limit would probably neglect the effects of higher intrinsic time scales, while an excessively high number would unnecessarily slow the computational time (as PE values are computed iteratively at each tau). After careful consideration, we decided to proceed with 100 as the maximum delay to apply on our resting-state hd-EEG data, based on two considerations: (i) the range of INT values shown in previous studies were comparable to those empirically observed at this threshold; and (ii) PE values do not considerably fluctuate well before our threshold. Embedding dimension *D* was set to 5 o balance the computational speed and the higher prediction accuracy, which is usually accomplished with higher *D* values [33].

In this study, we used the algorithmic implementation shown in [34] to calculate PE values, providing a fast computation time without sacrificing accuracy.

An example of the PE vs. *tau* graph and its associated minimum in an EEG channel is provided in Figure 1a.

### 2.2. ACW-0

The auto-correlation window (ACW) is an established method used to probe INTs [3,8,9]. It is based on the autocorrelation function *r*, which is defined as a signal *x*’s correlation with itself on different time lags:$$r_{l}=\frac{c_{l}}{c_{0}}\;c_{l}=\frac{1}{N}\overset{T-l}{\underset{t=1}{\sum }}(x_{\tau }-\bar{x})\left(x_{\tau +l}-\bar{x}\right)$$
where *l* denotes lag, and $\bar{x}$ denotes the mean of *x*.

Here, we computed the ACW-0, which is defined as the first zero-crossing of the temporal auto-correlation function (ACF) of the EEG time series [11]; ACW-0 might also be understood as the time lag after which the ACF crosses its 0% value. To this end, we computed a temporal autocorrelation with a lag of 0.5 with sliding windows of 20 s and a 50% overlap (10 s step size). Further details about ACW can be found in [3,9,14].

### 2.3. Simulations

Nonstationarity refers to a general property of signals with statistical moments (usually the mean or variance) that are not constant in time but vary to a certain degree; it is a property that biological signals, especially those recorded from the brain, display consistently [30].

Nonstationarity potentially affects the accuracy of time delay estimation. In [35], a simple method to assess the impact of nonstationarity on time-delay estimation with the use of synthetic signals is provided, building on a process originally described in [36] for financial time series. In this study, we have adapted the aforementioned method to test for the behavior of PE-TD in regimes of nonstationarity in a neuroscientific context: to this end, we integrate the process illustrated in [36] with a biologically plausible neuronal model, such as integrate-and-fire (IAF) models, to generate a simple case of in silico nonstationarity in the firing pattern of a single neuron. Integrate-and-fire neuron models [37,38] describe the behavior of a single neuron from the point of view of its membrane potential. An IAF neuron possesses its own resting membrane potential V_rest_: after it receives a series of inputs (inhibitory or excitatory), which can be modeled separately, it produces a spike only after its membrane potential surpasses a threshold V_th_. The potential is reset immediately after a spike is produced.

Parameters used for the IAF models used in this study are listed in Table 1.

The procedure can be broken down into a few consecutive steps.
First, we generated a series of “stationary” IAF time series of equal length (40,000 time points);Then, we obtained a nonstationary IAF signal by concatenating the previously synthetized stationary signals into a single signal;Eventually, by comparing the time scales measured in the synthetic nonstationary signal and the average of the time scale estimated on the stationary segments, one can assess the resilience of the tested measure to nonstationarity.

The rationale is rather straightforward: since the temporal structure of the concatenated nonstationary signal depends on the contribution of the temporal structure of its composing segments, a small distance between the time scales extracted from these two indicates high resistance of the tested measure.

Eight IAF segments of 4 s (40,000 time points) were generated with the parameters illustrated in Table 1. The input structure fed into the model varied randomly between each iteration to avoid repeating a particular stationary regime: inputs were randomly—but equally—distributed into either a DC input component (mean = 4 A, std = 1) or white noise with fixed mean (mean = 0) and variance = 1, effectively creating a simple case of a plausible nonstationary neuron: the input structure is known to affect IAF firing rate [39] and therefore its statistical properties. To compare signals of comparable length, we truncated the original stationary segments before concatenating them into the new nonstationary signal; in this way, we obtained a nonstationary signal of equal length (4 s). The procedure was repeated 500 times to avoid spurious results caused by sampling.

To characterize PE-TD’s behavior in a non-linear dynamic system model, we carried a parameter search over a very well-known model for non-linear feedback systems with intrinsic time delays: the Mackey–Glass oscillator [40]. In a Mackey–Glass system, a variable of interest x is under the control of a feedback system that, like in many biological systems [41], acts within a certain time lag, creating the conditions for the emergence of intrinsic time scales on its time series. In this study, the models were implemented by the following differential equation described in [21]:$$dx/dt=-x+\frac{ax\left(t-\tau_{s}\right)}{1+x^{c}\left(t-\tau_{s}\right)}$$
with *t* as a time index, $\tau_{s}$ as the time delay feedback, *a* as the feedback strength, and *c* describing the degree of nonlinearity. The Mackey–Glass differential equation was integrated via Euler’s method with time step Δt = 0.001; simulations lasted 5 s. A parameter search with a fixed *c* over a and tau was conducted by iterating one of the two parameters while fixing the other over the following range of values: *a* = 1 to 46 and *tau* = 50 to 300 with steps of 1.

### 2.4. Experimental Data (EEG)

Eighty-one participants with DoCs (39 UWS and 42 MCS; mean age = 46.65 ± 15.89 years old; sex-ratio = 2.24; etiology: stroke = 43; anoxia = 7; traumatic brain injury = 31) were recorded in resting state—at bedside—for 5 min using a Geodesics system (Ges300, EGI, Eugene, OR, USA) and a 256-channel electrode cap (HydroCel 130, EGI, Eugene, OR, USA) (following 10–20 international systems). Before the EEG recording session, the experimenters performed standard systematic procedures, such as the Arousal Facilitation Protocol [42], to ensure high wakefulness and arousal levels in the participants. No sedative agents were administered in the 24-h period that preceded the recording session to avoid drug-induced interference in the spontaneous brain activity’s signal. The severity of the disturbance of consciousness was assessed by administering, on admission, the Glasgow Coma Scale (GCS) [43], while the differential diagnosis was performed by trained clinicians by repeated behavioral assessments using the JFK Coma Recovery Scale–Revised (CRS-R) [42]. Additionally, a control sample of 44 healthy participants (age ± years) underwent a 5-min resting-state hd-EEG recording session; an additional sample of 20 healthy participants (age ± years) was used for further validation. The same aforementioned 256-channel system (GES 300, Electrical Geodesics, Inc., USA) was used to record both the datasets from the healthy participants. Healthy participants were asked to lie on the bed and keep their eyes open to mimic the experience of EEG recording in DoC patients. EEG data were re-referenced online to Cz and acquired at a sampling rate of 1000 Hz, while the impedance of all electrodes was kept to less than 20 KΩ. Further details about the datasets used in this study can be found in [14,24].

### 2.5. Pre-Processing

Pre-processing and data analysis, including statistical analysis, were carried out using in-house MATLAB software (The MathWorks, 2019b) and the EEGLAB toolbox [44].

The same pre-processing pipeline was used on all of the EEG datasets used in this study. First, the data were resampled to 250 Hz; then, a band-pass finite impulse response (FIR) filter between 0.5 and 40 Hz (Hamming window) was applied to the EEG channel data. Noisy channels were identified and rejected through a semi-automatic procedure. The rejection criteria used in our procedure were: removal of flatline channels (channels inactive for more than 5 s); correlated channels (with a correlation threshold of 0.8); low-frequency drifts; noisy channels; and short-timed bursts not related to neural activity (threshold at sd = 5 for data portions, relative to baseline). Next, bad channels were interpolated with a spherical method, and channel activity was re-referenced to the common average reference.

Artifacts related to eye movements, muscular noise, and heart activity were identified by independent component analysis (ICA), and their related independent components were removed from the signal.

### 2.6. Statistics

Root mean square error (RMSE) was used to assess the performance of PE-TD under nonstationary regimes in a series of LIF simulations.

To test for significant differences in the PE-TD values of the HC and UWS samples, Wilcoxon’s non-parametric rank-sum test was used, with the threshold level for the rejection of the null hypothesis set to 5%.

Spearman’s rank correlation coefficient was assessed to characterize the channel-wise relation between ACW-0 and PE-TD in both the conscious and unconscious groups. Furthermore, a bootstrap distribution test, with 10,000 iterations, was performed to assess the significance in the difference between Spearman’s correlation coefficients in conscious/unconscious states. The threshold for rejecting the null hypothesis was set to 5%. To further validate the state-dependent differences between correlation coefficients in different conditions, we assessed significance with Fisher’s z transform [45] as well.

## 3. Results

### 3.1. Simulations

In this section, we use synthetic signals to show how the estimation of time delays from time series data through PE-TD might be affected by: (i) nonstationarity; and (ii) parameter choice of model non-linear systems.

#### 3.1.1. The Effect of Nonstationarity on the Accuracy of PE-TD

An important a priori condition, when assessing the impact of nonstationarity on time delay estimation with the methodology described in Section 2.1, goes beyond the sheer comparison of stationary vs. nonstationary regimes. In fact, it is important to know that the model’s true time scale—the ground truth—can be recovered through the tested measure in the first place; if this condition is not met, the risk of spurious positive observations is not negligible. For what concerns an IAF neuron model, PE-TD is surprisingly accurate at recovering the average time delay between one spike and another, which involves simple recovery of time intervals from the spike rate (RMSE = 0.00015 s); this fact suggests that employing an IAF model is an appropriate choice.

A series of IAF stationary segments (see Figure 1b for an example) were concatenated into a single nonstationary signal (see Figure 1c for an example of such a signal) for 500 iterations to produce the testing substrate. The average error between the average PE-TD values obtained on stationary segments and on the nonstationary concatenated signals was very low (RMSE = 0.0011 s), suggesting that the impact of nonstationarity for the estimation of IAF time scales was minimal to null—at least for the time scales related to spike production in simplified model neurons.

#### 3.1.2. PE-TD Behavior as a Function of Parametrization Choice in a Non-Linear Delay System

As a first step, we evaluated the optimal parameter choice for c (the degree of non-linearity) in our Mackey–Glass oscillator for both auto-correlation window 0 (ACW-0) and PE-TD; we proceeded in this order since c was not a variable of interest in this study.

Figure 2a shows the error (computed as a simple algebraic subtraction) between the model’s time scale—which was fixed at tau = 160—and the time scale estimated with ACW-0, plotted against the parameter c; Figure 2b shows the same plot for PE-TD. For both measures, qualitatively similar results indicate that, for low c values, the estimation is quite unstable; however, beyond c = 10, the error becomes negligible, with an optimum for both measures reached at c = 16. However, PE-TD shows a peculiar steady decrease in accuracy after the dip reached at c = 16. Therefore, c was fixed at this value for the next round of simulations.

In the subsequent part of the study, we engaged in a parameter search—with a fixed c—over two parameters of interest, which are known to influence the state in which the Mackey–Glass oscillator is found [21]: the time scale tau and the feedback strength a.

Figure 2c shows the time scale estimation error plotted as a function of increasing tau (shown on the x axis) and a (shown as different colored lines) for ACW-0. With increasing values of tau, ACW-0 performance reaches stable accuracy only after the time delay feedback approaches tau = 160; this point is true for most feedback strength values beyond a = 6. We observed a similar pattern for feedback strength: at very low a values, it takes longer taus for the estimation to become more accurate compared with higher values.

PE-TD shows a similar behavior, as shown in Figure 2d. In fact, regardless of feedback strength, PE-TD seems to perform accurately when tau reaches a threshold of tau = 100. At shorter time scales, feedback strength seems to influence whether PE-TD overestimates the real underlying time scale.

### 3.2. PE-TD in EEG and DoC

#### 3.2.1. PE-TD Values Converge with INTs Probed with ACW-0

To be able to confirm the appropriateness of PE-TD for testing hypotheses relevant to the brain, we need to assess the degree to which this measure convergences and/or diverges with ACW-0, which has reinforced its position as the benchmark measure of INTs.

One of the most recognizable features of INTs is their hierarchical spatial organization: unimodal regions show shorter timescales compared to the longer time scales of transmodal cortical areas, related to the different time requirements to integrate (and segregate) their perceptual space [3]. Do findings obtained through PE-TD converge with this important property of the brain? The topoplot in Figure 3a shows (although in electrode space) a fair hierarchical distribution of PE-TD values, coherent with findings obtained with ACW-0. We observed a significantly high spatial correlation between the two maps obtained with PE-TD and ACW-0 (Figure 4a, R = 0.90; *p* < 0.001), suggesting a preservation of the hierarchical spatial distribution of INTs captured by PE-TD. To validate these findings, we repeated the same analysis on another sample of healthy subjects (N = 20) with similar results (R = 0.93, *p* < 0.001).

Previous studies [5,14,24] have shown that loss of consciousness is accompanied by an abnormal prolongation of INTs, as probed with ACW-0. Does this outcome apply to methods that estimate INTs through information theory? Replicating previous results, PE-TD values in both UWS and MCS showed a significantly higher subject-wise average compared to those from healthy controls, but there were no significant differences between UWS and MCS (Figure 3c. HC mean = 0.19 s; UWS mean = 0.24 s; MCS mean = 0.27 s; HC vs. UWS: *p* < 0.001; HC vs. MCS: *p* < 0.001; UWS vs. MCS: *p* > 0.05). To summarize, our findings showed that, through PE-TD, we were able to replicate: (i) the hierarchical distribution of INTs in two unrelated healthy populations; and (ii) the abnormal prolongation of INTs in resting-state EEG recordings in DoC patients previously shown in other studies, with fairly similar data distributions.

#### 3.2.2. Increased Distance between INTs Obtained with PE-TD and ACW-0 during Loss of Consciousness

Loss of consciousness not only entails a different capacity for input processing [28] but also a significant change in signal properties [46]. Is it possible to take advantage of these differences to qualify the differences between conscious and unconscious subjects? We introduce a characterization of loss of consciousness as an increase in the discrepancy between the spatial organization of INTs probed with PE-TD and that obtained with ACW-0.

To this end, we used the channel-wise linear correlation between average values of PE-TD and ACW-0 as a proxy measure of the spatial similarity between the two measures; then, we tested for statistical differences between the correlation coefficients obtained with HC and UWS. Figure 4 shows that the linear relation between PE-TD and ACW-0 changes drastically from HC (Figure 4a) to UWS (Figure 4b) (R_HC_ = 0.90; *p* < 0.001; R_UWS_ = 0.55; *p* < 0.001). We tested for significance through the bootstrap distribution test described in Section 2.4. The difference in the correlation coefficients was highly significant (*p* < 0.001): the statistical significance was confirmed and validated with Fisher’s z transformation test (*p* < 0.001).

## 4. Discussion

The purpose of this study was to show that the estimation of a system’s time delays through serial computation of permutation entropy (PE) values as a function of its embedding time delay parameter (which we refer to as PE-TD, permutation entropy–time delay estimation) is compatible with the estimation of neural activity’s time scales; additionally, we suggest a parallel utilization of PE-TD, which could offer some important insights into the mechanisms of consciousness as well, coherent with the theoretical framework of TTC.

With this point in mind, we followed two parallel strategies. In the first part of the study, through a series of simulations, we demonstrated that PE-TD works well in estimating the time scale of a simple IAF neuron model and that it is well resistant to nonstationary regimes; furthermore, we showed how the performance of PE-TD in a nonlinear time-delayed model, such as the Mackey–Glass oscillator, is affected by parametrization choice, which could hint at strategies aimed to improve the interpretation of PE-TD in real-world applications.

In the second part, we employed PE-TD on a dataset consisting of resting-state hd-EEG recordings of DoC patients and healthy controls, demonstrating that PE-TD is able to replicate the abnormal prolongation of INTs during clinical loss of consciousness, as previously shown in other studies; additionally, we observed a divergence in the results obtained through PE-TD and ACW-0 when consciousness is lost, which we suggest is due to different signal properties of EEG recordings of people with DoCs.

The estimation of INTs is a fundamental aspect of TTC, which posits an important role for the temporal structure of the brain’s spontaneous activity in shaping consciousness. A crucial assumption for this postulate is that the brain has adapted its features to the external environment’s statistical input structure, an evolutionary force driven by the limited computational resources of the brain and the consequential need to maximize the information extracted from the outside world [47]: the dynamics of its intrinsic temporal structure (e.g., INTs) make no exception because inputs/stimuli possess their own temporal scales (e.g., the temporal dimension at which the input structure changes its relevant features), and the brain “aligns” to these timescales for proper and efficient encoding.

Additionally, a peculiar characteristic of TTC is the notion of “common currencies” [48]: e.g., the dimensions of consciousness and neural activity share topographical and dynamical intrinsic properties. Hence, the utmost importance of estimating INTs in this theoretical framework encourages methodological advances to ensure the soundness of the scientific approach to the study of consciousness.

In this study, we observed that our first implementation of PE-TD in a neuroscientific context worked surprisingly well in extracting the average time between two consecutive spikes of an IAF neuron directly from its time series; therefore, we suggest that PE-TD is fit to study neural time scales even at a microscale level of investigation (e.g., single neurons).

Other studies have demonstrated that PE does not require a strict assumption of stationarity of the underlying process [49]: this fact is reflected in the overall better performance of PE as an estimator of financial time series, as recently shown in [35]. In this study, we present similar evidence as in [35] but within the context of a single spiking neuron, which helps to contextualize this informational theoretic approach to a different field of knowledge. Furthermore, we linked the potential cause of nonstationarity in a neural time series (a varying external input structure) to its effects on the behavior of a single spiking neuron.

In cases such as delay systems, non-linear effects are non-negligible [1]. For this reason, a measure capable of extracting the underlying temporal structure of the system of interest with sufficient accuracy is of utmost importance.

We were interested in unveiling the probable causes of the differences between ACW—which is the standard measure for estimating time delays in neural signals—and PE-TD. Computing the ACW of a signal requires the computation of its autocorrelation function (ACF). ACF (and therefore ACW) is known to behave similarly to ordinal-based time scale quantifiers in linear conditions [35] and is actually a better solution when it comes to quantify linear dynamics [50]. However, most real-world systems—including the brain—have non-linear dynamics: as such, it is important to know the basis of divergence between these two measures.

First, we established the minimum degree of non-linearity of a Mackey–Glass oscillator (c coefficient, see Section 2—Methods) that resulted in comparable performance between ACW-0 and PE-TD.

Second, we observed different behaviors of ACW-0 and PE-TD when both models’ timescales and the feedback strength varied. For ACW-0, the estimation error seems to decrease monotonically as the model’s intrinsic timescale becomes longer, achieving stable performance after a threshold (tau = 160). Accordingly, for the feedback strength parameter, we show an unsurprising trend toward higher accuracy with higher feedback values, which however can be appreciated only at shorter timescales. If similar results were to be shown for neural simulated data, it would suggest that neural populations with sufficiently strong reentrant connections and with sufficiently long intrinsic time signatures can be efficiently probed using ACW-0 with an acceptable estimation error.

On the other hand, PE-TD displayed a different but comparable performance. The lower error range suggests higher accuracy of PE-TD for the estimation of timescales in the Mackey–Glass model; furthermore, the accuracy reaches stability at shorter values of tau, suggesting that PE-TD is a better fit when we assume shorter intrinsic timescales in the neural population of interest. With respect to feedback, we did not observe any meaningful difference with the performance of ACW-0. Our results are useful to informing real-world application of these two different time delay quantifiers; however, we do not want to advance the claim that these observations can translate directly to neuroscience. Further simulation studies will need to quantify the minimum values of connection strength and the durations of intrinsic dynamics that are acceptable for a correct time delay estimation.

While our simulations provide a first set of evidence that advances the use of PE-TD in a neuroscientific context, this study also advances the use of PE-TD on real-world neural data by replicating a series of results obtained from hd-EEG datasets that have already been probed with ACW-0 in past studies [14,24].

We confirm that PE-TD is able to satisfactorily reveal the established INT hierarchical distribution across the scalp [11], with the high correlation observed at the channel level, between INTs measured by ACW-0 and PE-TD (Figure 4a). To further validate this claim, we confirmed the same results in a second independent dataset. A postero-anterior gradient of intrinsic timescales can be qualitatively appreciated by the two very similar topographical plot (topoplots) shown in Figure 3a,b.

Moreover, we also replicated the same abnormal prolongation of INTs that is typically appreciated in different unconscious states [14,24], as in the case of disorders of consciousness (DoCs) (Figure 3c). As an additional confirmation of the validity of these results, replication was not restricted to the aforementioned abnormal prolongation of INTs: we also observed a similar range of INT values compared to those obtained through ACW-0. Together, our results encourage the use of PE-TD in the investigation of human INTs as well since it produces outputs comparable to ACW-0.

Altogether, these findings suggest a close, but not complete, overlap between the timescales estimated by PE-TD and ACW-0. This parallelism allows for an indirect link between PE-TD and the spectral characteristics of an EEG signal (e.g., its power spectrum density, PSD). In fact, despite a well-known mathematical relation between a signal’s ACF and its PSD, there is evidence of a clear dissociation between the timescale probed in a signal and its spectral features [14], suggesting that the correspondence between the two is far from trivial: for instance, two signals can show a similar PSD but very different ACW values, and vice versa, two signals with the same timescales might have very different PSDs. In the case of PE-TD, even if there is no direct link between the spectral composition of the brain signals and their estimated timescales, it is plausible to infer a similar degree of dissociation: even if a slower spectrum (e.g., with higher power in the slow frequency range) is expected to produce slower INT values on average, the relation between the two is likely nonlinear due to the contribution of the aperiodic component present in neural signals. Future studies will need to shed light on this conundrum—especially in light of the availability of the novel methodology presented in this study. In the last section of this study, we advanced one step further and took advantage of the inherent differences between ACW and PE-TD as an instrument to gain further insight into the neurophysiological features of consciousness.

The channel-wise correlation between ACW-0 and PE-TD can be thought of as a “closeness” measure: the higher that the correlation is, the more similar that the results obtained and their spatial distribution across the scalp are, while lower correlation values would signal increased distance between the two. With this idea in mind, we asked whether the “closeness” between ACW-0 and PE-TD would show a state-dependent difference when consciousness is severely impaired, as in unresponsive wakefulness syndrome (UWS). We observed a significant drop in the correlation coefficient in UWS compared to the healthy controls (HC) sample, even if the correlation coefficient remained moderate (but nonetheless significant).

How can one interpret this increased distance between ACW-0 and PE-TD when consciousness is lost or impaired? We suggest that the cause must be searched for in the different signal properties related to loss of consciousness. For instance, unconscious states have recently been characterized by a qualitatively distinct nonstationary attractor landscape [29], which constrains the past and future states of a system. Therefore, differences in the behavior of a dynamical system are expected to generate differences in the properties of its time series, for instance, its nonstationarity. PE, as we have demonstrated, is resilient to the signal’s nonstationarities; this fact suggests that the increased distance between the time scales obtained with PE-TD and those obtained with another method, such as the ACW, might be due to increased differences in the signal’s nonstationary characteristics. We suggest that this characterization of different states of consciousness relates to the nonlinear behavior of a system and its statistical features (e.g., the degree of nonstationarity in its time series). This theory is consistent with many current theories of consciousness: for instance, the degree to which brain activity is differentiated in time is a relevant indicator of consciousness in the latest iterations of IIT [51], while the dynamic properties of the broadcasting networks are coherent with increasing nonstationarity during conscious access [52]. In the framework of TTC, these findings are also coherent with an expected poorer “dynamic repertoire” [53] of the spontaneous activity’s temporospatial structure characterizing unconscious states [27,28]. Indeed, TTC advances the claim that consciousness does not only manifest in the time series data captured from the brain as neural correlates of consciousness (NCCs, [51]), but it is also dependent on the preservation of baseline conditions that are necessary—but not sufficient—for the actuation of consciousness, summarized in the notion of neural predisposing factors of consciousness (NPCs) [28,54]. For instance, in this theoretical framework, NPC candidates are represented by the “scale-freeness” of the brain’s spontaneous activity [14,55] and the richness of its intrinsic dynamics [31,32,56], which are known properties of nonlinear physical systems: a shift (or disruption) in these properties is expected to drive unexpected consequences for their signals’ statistical properties [57]. Therefore, this disruption of nonlinear properties in the brain’s spontaneous activity and its effects on the statistical features of the time series that it generates are tied to potential differences in nonstationarity regimes between conscious and unconscious states, explaining the ACW-PE-TD distance that we observed in UWS. However, this theory requires strong experimental evidence; thus, future studies are warranted to investigate this particular interplay between these different TTC-based mechanisms/dimensions of consciousness.

### Limitations


Thus far, we have provided evidence for the use of ordinal quantifiers (PE) to estimate neural time scales. However, a recent study [33] showed that the use of ordinal statistics still has room for improvement, e.g., using weighted permutation entropy (PE) to account for amplitude. In this study—which to our knowledge is the first neuroscientific example of the use of ordinal quantifiers for time-delay estimation—we proceeded with PE because of how better understood it is in comparison with its more recent variants.PE-TD, in its first implementation, only takes the absolute minimum of the PE vs. time delay graph as its estimated INT. We followed these heuristics of the novelty of this approach in neuroscience and, furthermore, to allow for an approachable comparison with the ACW. However, we do not suggest that the absolute minimum is always the relevant time scale of an EEG signal, as multiple time scales are to be expected, even in the same brain population: this fact is already implied in the way that INTs are extracted through ACW, which has multiple versions [8] that are believed to capture different neural time scales. Pragmatically, the high correlation between PE-TD and ACW-0 in our healthy subjects suggests that, at least for the preliminary use of PE-TD, it is an appropriate choice; however, future studies must include the investigation of other local minima for the relation of different minima to neural/behavioral events.On the other hand, along with the replication of the abnormally prolonged average INT values in DoCs obtained through ACW [14,24] we observe a similar lack of significance in the difference between the UWS and MCS diagnostic groups, as already shown in [24]. Because of the current clinical challenge posed by the presence of covert consciousness [58,59], we argue that the lack of predictive power of average INT values to distinguish between different states of consciousness, even with PE-TD, is not an intrinsic weakness of this approach but rather a symptom of the discrepancy between behavioral responsiveness and consciousness itself [26]. Therefore, we encourage further studies, which are needed to refine the study of INTs and to improve on its potential as a diagnostic/prognostic marker of conscious states.


## 5. Conclusions

Intrinsic neural timescales (INTs) are a remarkable feature of the human brain: they are related to temporal input processing and, in the theoretical framework of TTC, allow for adequate conscious states through their interaction with the environment—namely “temporospatial alignment”. However, INTs are not known a priori and need to be estimated from neurophysiological data. In this paper, we have advanced the use of permutation entropy (PE) to this end, a methodology that is already applied to estimate time delays in different fields of physics. First, we tested the suitability of this methodology for its use on neural data with synthetic time series, demonstrating its resistance to extreme regimes of nonstationarity and providing some heuristics for interpreting output values. Moreover, our empirical investigation motivates the use of PE-TD in hd-EEG data, replicating previous results and yielding high similarity with values obtained with previous standard methodologies (ACW-0). Further, we also observed an increased dissimilarity between INTs probed with PE-TD and with ACW-0 in clinical loss of consciousness: we suggest that this finding is a first step toward a deeper characterization of different states of consciousness. In conclusion, we demonstrated that PE-TD is a valid methodology for the measurement of INTs from resting-state EEG data and we further propose that, because of its characteristic resistance to nonstationarity, it could be even helpful to better discriminate between different state of consciousness.

## Figures and Tables

**Figure 1 entropy-25-01086-f001:**
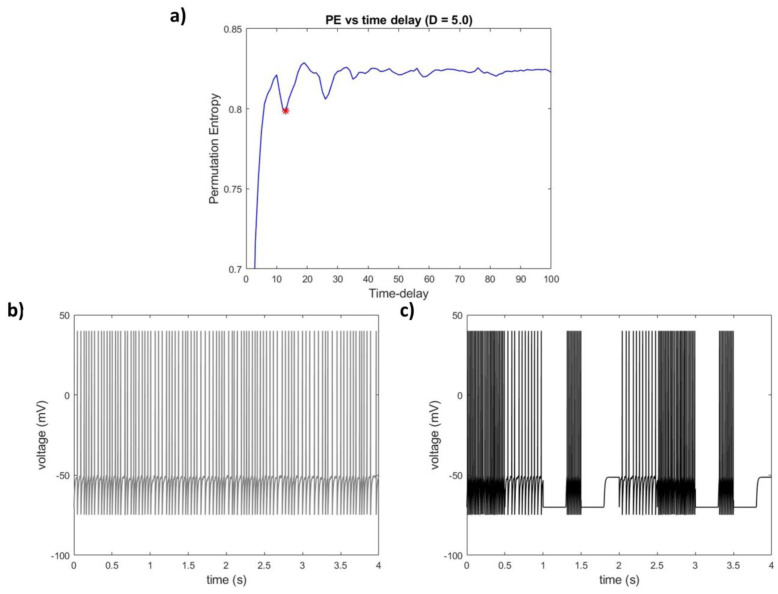
Time delay estimation through permutation entropy (PE) in a neural environment. (**a**) PE values as a function of increasing time embedding tau for a single channel hd-EEG recording. Estimating time delays in a physical system through PE relies on the notion that, when the tau parameter matches the time delay of the system, PE values are expected to dip significantly. The distribution of permutation patterns obtained with a tau matching the system’s dominating temporal scale is narrower; therefore, the system becomes more “predictable”: thus the dip in PE values. (Red star indicates the minimum of the PE vs time-delay graph.) (**b**) An example of a stationary integrate and fire (IAF) neuron signal. (**c**) Concatenating multiple (8) IAF stationary segments to obtain a simple case of a synthetic nonstationary signal allowed us to investigate the effects of nonstationarity on PE-TD.

**Figure 2 entropy-25-01086-f002:**
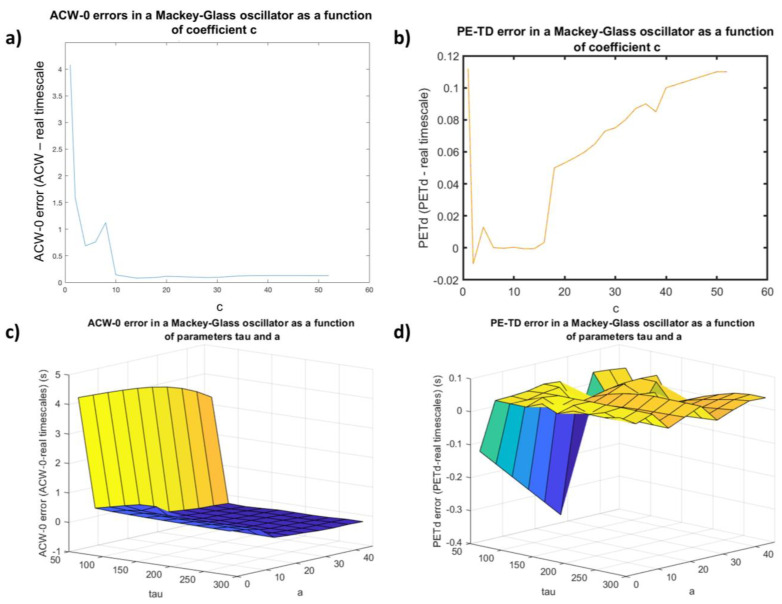
Differential behavior of autocorrelation window (ACW-0) and PE-TD in model parameters for a model nonlinear delay system to perform the interpretation of estimated time delays. (**a**) Estimation error (in seconds) as a function of parameter c (degree of nonlinearity) when using ACW-0 to estimate the time delay of a Mackey–Glass oscillator. Stable performance is reached at c = 16. (**b**) Same graph for PE-TD. PE-TD behaves similarly to ACW-0, as the performance reaches an optimal accuracy around c = 16. However, after this increase in accuracy, the performance of PE-TD decreases steadily as a function of c. (**c**) Third plot of the estimation error as a function of both a (feedback strength) and tau (time delay) when using ACW-0. (**d**) Same graph for PE-TD. PE-TD seems to perform stably at earlier timescales and with weaker feedback strength than ACW-0.

**Figure 3 entropy-25-01086-f003:**
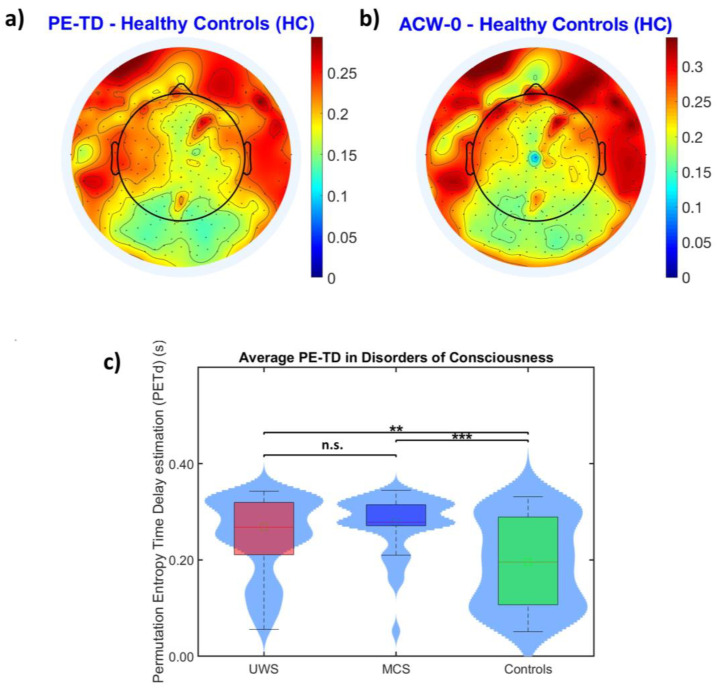
The distribution of INTs in healthy populations and their abnormal average prolongation in disorders of consciousness (DoCs), probed with PE-TD. (**a**) Topopolot depicting the average distribution of INT values (in seconds) probed with PE-TD in a healthy population (N = 44). (**b**) Topoplot depicting the same average distribution of INT values probed with ACW-0. The overall INT scalp distribution is clearly consistent between the two different measures, as confirmed by a very high channel-wise correlation between the two measures (R = 0.90, *p* < 0.001, presented in Figure 4). (**c**) Violin plots for the subject average length of PE-TD values in DoCs vs. healthy controls (HC). (HC mean = 0.19 s; UWS mean = 0.24 s; MCS mean = 0.27 s; HC vs. UWS: *p* < 0.001; HC vs. MCS: *p* < 0.001; UWS vs. MCS: *p* > 0.05). (** represents *p* < 0.01 and *** represents *p* < 0.001. n.s., when shown, stands for “non-significant” (*p* > 0.05)).

**Figure 4 entropy-25-01086-f004:**
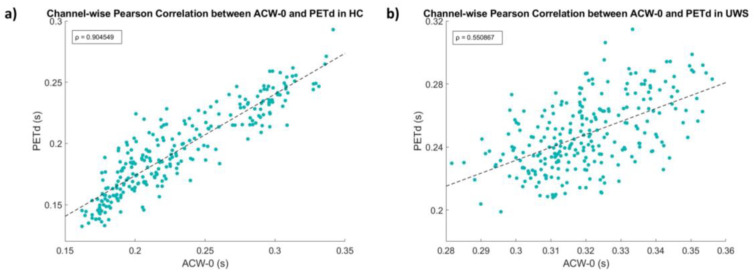
State-dependent decreased channel-wise correlation between PE-TD and ACW-0 in loss of consciousness (LOC). (**a**) Pearson’s correlation coefficient in healthy subjects (R = 0.90, *p* < 0.001). (**b**) Pearson’s correlation coefficient in UWS (R = 0.55, *p* < 0.001). The decrease in correlation observed in UWS is confirmed by a bootstrap distribution test (*p* < 0.001) and is validated with a Fisher’s z transform test (*p* < 0.001) (see Section 2—Methods).

**Table 1 entropy-25-01086-t001:** Model parameters for the integrate-and-fire (IAF) neuron used in this study.

Parameter	Value
V_rest_	−70 mV
V_reset_	−75 mV
V_th_	−50 mV
Sampling rate	10 kHz
Resistance	10 MΩs
Decay time constant	10 ms

## Data Availability

Data used in this article are subjected to sharing restrictions due to privacy issues regarding sensitive clinical data. MATLAB (R2019a release) was used for this study. Most of the data analysis was conducted using the EEGLAB (http://sccn.ucsd.edu/eeglab/ (accessed on 6 June 2023)) toolbox, an open-source MATLAB package. The custom MATLAB scripts used in this study are available upon reasonable request. Relevant code to replicate our analysis is freely available at http://www.georgnorthoff.com/code (accessed on 6 June 2023).

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
