# Peer review of "Probing Intrinsic Neural Timescales in EEG with an Information-Theory Inspired Approach: Permutation Entropy Time Delay Estimation (PE-TD)"

_entropy, 2023, doi:10.3390/e25071086_

Round 1
Reviewer 1 Report
The team introduces a new method to measure a particular neural time scale (it uses the entropy concept, so it would be appropriate for this journal), They find that the measure is strongly correlated to another well-known method, so it does not appear to improve on existing methods. But then the team shows that this correlation is much weaker in the EEG signal from patients that are losing consciousness. They argue from that observation that perhaps their method can help characterize conscious states. This latter conclusion seems to me to be far-fetched. While the evidence is clear that the two methods do not correlate so well for the loss of consciousness data, this does not imply that the PE-TD method is the one that can characterize loss of consciousness better. It is also not clear (to me or to the authors) for what reason TE-PD should have an advantage in characterizing consciousness states. For me, that entire section is highly speculative, but unfortunately it is also the only one where the new method behaves differently from the older one (called ACW). All in all, there is novelty in this paper, as it seems that the PE-TD method (which in itself appears sound to me) has not been used on neurological data before. It is also appropriate to first start with synthetic data, as the authors did. Perhaps a suggestion is to try to model loss of consciousness in synthetic data, so as to pinpoint the difference between PE-TD and ACW in this domain. This might reinforce the last section.The English is at times somewhat tortured, but it is understandable throughout.
Author Response
Dear reviewer,
thank you for your comments.
Let us elaborate in detail our position regarding your suggestions.
- "[...] it is also not clear (to me or to the authors) for what reason TE-PD should have an advantage in characterizing consciousness states. For me, that entire section is highly speculative, but unfortunately it is also the only one where the new method behaves differently from the older one (called ACW). "
We totally agree when you mention the fact that PE-TD values by themselves - or even the degree of spatial similarity between ACW-0 and PE-TD values - are not sufficient to characterize the full extent of conscious states. However, we do imply several times throughout the paper, and especially in the last paragraph of the Conclusion section (lines 582-585) that it's PE-TD's resistance to nonstationarity, and its behavior in the presence of nonlinearity, that makes it more suited to estimate INT values with more accuracy: since we expect differences in nonstationarity and nonlinearity between conscious and unconscious states, a measure that is more robust despite changes in these properties is a better candidate for a contrast between consciousness/unconsciousness. In fact, the decreased similarity between PE-TD and ACW-0 INT maps is, in our opinion, only an epiphenomenon which we suggest reveals, in concordance with previous literature (which is mentioned in the last paragraphs of the Discussion section, lines 518-544), that loss of consciousness entails different properties of the recorded neural activity, including nonstationarity and differences in nonlinearity.
- " Perhaps a suggestion is to try to model loss of consciousness in synthetic data, so as to pinpoint the difference between PE-TD and ACW in this domain. This might reinforce the last section."
This last point is very important to us, and we thank you for bringing it up. Modeling loss of consciousness in order to reveal whether it is possible to replicate the same pattern observed in our findings (e.g. differences in nonstationarity lead to an increased distance between the topographical INT maps revealed by ACW-0 and PE-TD) is of utmost importance, and we are actively pursuing this research question. However, as this involves finding the appropriate model to characterize loss of consciousness - which is a task far from trivial, and still actively debated in consciousness research - we think that it's out of scope for the sake of our article. We are currently chatacterizing a simple model which aims at investigating the exact same questions you arised. We completely agree that an in silico study as such could complete and reinforce our findings, but we think that an extensive and appropriate analysis would not fit as an additional section of this work, which we argue is already methodologically dense as it is now. For an extensive and recent review on the challenges posed by modeling loss of consciousness in silico, we suggest the following review: Luppi et al., 2023, Computational modelling in disorders of consciousness: Closing the gap towards personalised models for restoring consciousness, Neuroimage.
We hope you can understand your point of view: we also hope to present interesting findings soon regarding this matter, in order to extend our findings.
We appreciate the time you spent to read and carefully review our work, and we feel that your suggestion encourages us to go further in this investigation.
Reviewer 2 Report
In this manuscript, Buccallato and colleagues take the permutation entropy-based time delay estimation method and compare it to the commonly used Autocorrelation Window method to compare estimation of Intrinsic Neural Timescales. They first probe simulated data from IAF neuron models and from Mackey-Glass models to analyse the effect of increasing non-stationarity. Here the authors found that non-stationarity was not a big influence on PE-TD in the IAF signal, but had more of an influence in the Mackey-Glass model data. They then compare the methods in EEG recordings taken from patients suffering from disorders of consciousness, finding a less strong correlation between the permutation and autocorrelation measures in DOC patients compared to healthy controls.
My comments on the manuscript are as follows:
- In the abstract and introduction the TTC is given quite a bit of focus, but it is not really part of the methods presented nor of the results found. While the findings are coherent with the theory, they might also be coherent with many of the other theories of consciousness. I would suggest that the focus on TTC is toned down in the abstract and introduction, and shifted to the discussion.
- Some of the results sections are restatements of aims and methods (e.g., lines 371-377, 380-388, 395 - 399 etc.). The paper would be more readable if these sections were removed. The discussion also has sections largely restating methods (e.g., lines 462 – 467).
- I think it would be worth clarifying the relationship between INTs and classic spectral measures, at least in the discussion. For example, a stable INT seems to imply a periodicity, i.e. that an oscillation with some kind of comparatively stable frequency would be present. If an EEG signal is dominated by 1/f noise alone (as it can be in DOC patients), or shows clear burst suppression, what would an intrinsic neural timescale value then represent? If multiple periodicities are present in the signal, does an INT represent the slowest one?
- Some of the text in the figures is very small, please enlarge.
- Minor textual things:
- line 76 ‘over a signal that shows periodic patterns’
- line 78 ‘have successfully found coherent results’
- line 82 please fix reference, number form.
- line 107 consider ‘particular’ instead of ‘peculiar’
- line 203 is formatting missing in ‘and x denotes the mean of x.’?
- line 284 I suggest ‘carried out’ to replace ‘carried on’.
- In figure 4 are the values truly R-squared, or is it just rho? In the text it is just rho (r). Please add scale to y-axis.
- line 451, ‘works’ is repeated.
- line 458 instead of ‘contact’, ‘context’.
- line 514 the authors state “… even if the correlation coefficient remained relatively high”. Not many would agree that a rho of 0.55 is ‘relatively high’, please reconsider.
- line 519 ‘attractor landscape’ does not require an acronym.
I have added a few suggestions in my review text. In my opinion the manuscript could be made more concise for ease of reading.
Author Response
Dear reviewer,
thank you for your very useful comments and your interesting insights on our work. We would like to respond in detail to your comments, point by point, in the next paragraphs.
- "In the abstract and introduction the TTC is given quite a bit of focus, but it is not really part of the methods presented nor of the results found. While the findings are coherent with the theory, they might also be coherent with many of the other theories of consciousness. I would suggest that the focus on TTC is toned down in the abstract and introduction, and shifted to the discussion."
Thank you for this practical suggestion. After careful consideration, we have come to the conclusion that the link of our findings to TTC was indeed more appropriate for the Discussion section, as you rightfully suggested: for this reason, we shifted to focus on TTC to the Discussion section (lines 432-443), and toned down the implications of TTC throughout the article.
- "Some of the results sections are restatements of aims and methods (e.g., lines 371-377, 380-388, 395 - 399 etc.). The paper would be more readable if these sections were removed. The discussion also has sections largely restating methods (e.g., lines 462 – 467)."
We have shortened or removed when necessary restatements of aims and methods throughout the manuscript as suggested, and we agree that avoiding redundancy has improved the readibility of our work. Thank you for this important comment.
- "I think it would be worth clarifying the relationship between INTs and classic spectral measures, at least in the discussion. For example, a stable INT seems to imply a periodicity, i.e. that an oscillation with some kind of comparatively stable frequency would be present. If an EEG signal is dominated by 1/f noise alone (as it can be in DOC patients), or shows clear burst suppression, what would an intrinsic neural timescale value then represent? If multiple periodicities are present in the signal, does an INT represent the slowest one?"
Thank you for this interesting question, which has implications that are far from trivial. We have added an additional paragraph to elaborate more on that in the discussion section (lines 497 - 510), but we'd like to add some details here, in order to avoid an increase in the complexity of our manuscript. As said in the text, the relation between a signal's ACF and its PSD is well-established and almost tautological: the Wiener-Khinchin theorem implies that one can be derived from the other. However, the ACW-0 relies on ACF but is not completely equal to it, as it aims at finding the time at which the ACF decays to its 0% value. Therefore, the relation between ACW and its spectral features is not a linear one, and recent evidence (Zilio et al., 2021, cited in our manuscript) has produced evidence of that. Assuming that ACW-0 and PE-TD are good estimators of INTs, we suggest it's safe to think that the same non-linear relationship between PE-TD and the spectral properties of a signal exists. Moreover, INTs and more generally the estimation of time scales in time series produce by complex dynamical systems are not only tied to its oscillatory components, but also to its aperiodic ones (Beran et al. 2013, Long memory processes: probabilistic properties and statistical methods, Springer).
Additionally, we mention (in lines 534-539) that multiple time scales are expected in neural activity time series, and that reducing INT estimation to a single value is a choice of parametrization with both ACW and PE-TD. This effect can also be appreciated in figure 1, in the Permutation Entropy vs tau graph: multiple local minima are presented, but we chose the relatively lowest value of PE as it was a coherent choice with previous studies.
- we have implemented all suggested changes in the manuscript, including changes in the text and improvement in the figures.
We are grateful for your work. We hope to have answered with sufficient clarity to your comments.
Round 2
Reviewer 2 Report
I am happy with the changes the authors have made. Thanks to the authors also for the helpful comments regarding the relationship to spectral measures. All the best with the publication.